# Design and Implementation of a Chain-Type Direct Push Drilling Rig for Contaminated Sites

**DOI:** 10.3390/ijerph20043757

**Published:** 2023-02-20

**Authors:** Pinghe Sun, Shengwei Zhou, Han Cao, Guojun Cai, Shaohe Zhang, Qiang Gao, Gongbi Cheng, Biao Liu, Gongping Liu, Xinxin Zhang, Yun Liu, Dongyu Wu, Zhenyu Ding, Lan Zeng, Guangdong Liao, Leilei Liu, Xiaokang Wang, Ting Xiao, Jing Jin, Hanhan Yang

**Affiliations:** 1Key Laboratory of Metallogenic Prediction of Nonferrous Metals and Geological Environment Monitoring, Ministry of Education, Changsha 410083, China; 2School of Geosciences and Info-Physics, Central South University, Changsha 410083, China; 3School of Civil Engineering, Anhui Jianzhu University, Hefei 230601, China; 4Jiangsu Gaiya Environmental Science and Technology Co., Ltd., Suzhou 215000, China; 5College of Chemical Engineering, Nanjing Tech University, Nanjing 211189, China; 6Institute of Soil Science, Chinese Academy of Sciences, Nanjing 210008, China; 7Chinese Academy of Environmental Planning, Ministry of Environmental Protection of China, Beijing 100012, China; 8Suntime Environmental Remediation Co., Ltd., Changzhou 213000, China

**Keywords:** chain-type, direct push drilling rig, lightweight drilling rig, contaminated sites investigation

## Abstract

For sites where volatile organic compounds are present, the direct push method, in combination with other sensors for investigation, is a powerful method. The investigation process is an integrated drilling and sensing process, but the trajectory of the probe carrying the sensor is ambiguous. This paper explores and introduces the application of a chain-type direct push drilling rig by designing and building a chain-type direct push miniature drilling rig. This rig allows for indoor experimental studies of direct push trajectories. The chain-type direct push drilling model is proposed based on the mechanism of chain transmission. The drilling rig provides a steady direct thrust through the chain, which is driven by a hydraulic motor. In addition, the drilling tests and results described prove that the chain could be applied to direct push drilling. The chain-type direct push drilling rig can drill to a depth of 1940 mm in single-pass and up to 20,000 mm in multiple passes. The test results also indicate that it drills a total length of 462.461 mm and stops after 87.545 s of operation. The machine can provide a drilling angle of 0–90° and keep the borehole angle fluctuating within 0.6° with the characteristics of strong adjustability, flexibility, continuity, stability, and low disturbance, which is of great value and significance for studying the drilling trajectory of direct push tools and obtaining more accurate investigation data.

## 1. Introduction

As cities grow and more chemical factories move away, so do the number of contaminated sites and the types of contaminants. A contaminated site is one of the current study hotspots, such as the distribution characteristics [1,2,3], risk assessment [4,5], remediation [6,7,8,9,10], and impact on human health [11,12]. Due to the typical characteristics of hidden, irreversible, and complex contaminated sites in China [13], a contaminated sites investigation has received more attention. The purpose of contaminated sites investigations is to determine whether the soil in the site is contaminated and the degree of the contaminant. The best remediation of contaminated sites is determined by investigating the types and contents of contaminants and screening the remediation technologies [14].

Direct push logs were formally used in the 1930s for cone penetration testing for geotechnical site characterization. For a contaminated sites investigation, the equipment commonly used are the SH30 drilling rig and direct push GP drilling rig, in addition to the XY and DDP-100 drilling rigs (automobile drilling) [15]. Among them, Geoprobe’s direct push drilling rigs are widely used in sampling groundwater [16,17] in Germany, and they are also used for sampling sediments in the United States [18]. Moreover, a series of policy documents were enacted, and a lot of work was performed in China after the poisoning of construction workers in the Songjiazhuang subway. In previous work, direct push drilling was applied in the fields of contaminated sites investigations and geochemistry. For example, the Geoprobe series of direct push soil drilling rigs from the Kejr branch of Geoprobe Systems have been used at a large abandoned industrial and mining site in Beijing. In addition, a detailed investigation of the soil and water environment was performed by using static pressure and hammering waterless drilling techniques, respectively [19]. Contaminated sites are inevitably disturbed by sampling, resulting in different levels of transport and dispersion of their gaseous and liquid contaminants. Based on whether the chemical properties of the contaminants change when disturbed, contaminants in rock and soil mass are classified into stable contaminants and unstable contaminants by the American Material Testing Standard ASTM D6169 [20]. Therefore, the investigation of different types of contaminants requires different methods to ensure the least disturbance on the original soil [21].

The in-situ investigation on contaminated sites is an extremely important approach, which can avoid the high cost and time-consuming impact of a sampling investigation. Multiple investigation methods are combined with direct push drilling techniques to investigate contaminated sites. Techniques such as UVOST, Membrane Interface Probe (MIP), and Hydrodynamic Profiling Tool (HPT), which are emerging tools for the delineation of lithology and contamination, can improve the resolution of spatial data [22]. MIP was first introduced in 1996 as a tool for investigating VOCs and is now used in VOC detection in several countries [23,24,25]. The MIP logging tools are widely used to measure volatile petroleum hydrocarbon and solvent contamination, along with the electrical conductivity of soil and permeability. The most commonly deployed components include the MH6534 MIHPT probe, MIHPT trunkline (MN 236310)/MIP trunk (MN 202570), HPT pressure sensor, connection section, drive head and probe rods, depending on the size of the rod string being driven into the ground, and the lithology or permeability sensors used in combination with the MIP. Successive sections of these rods are added to push or percussion drive the probe to certain depths. At present, MIP can achieve uniform penetration into the subsurface at a certain speed and perform VOC measurements. The direct push drilling process and the determination of VOCs can be realized as a mature continuous process. The MIP and HPT were combined for the subsurface investigation of unconsolidated formations, and this method can obtain both VOC and relative formation permeability data, and the results of investigation show that this method has a better performance [26]. The current MIP-IN technology combines MIP technology with the direct-push injection of the remediation agent to reduce remediation time and costs [27,28]. In addition to MIP technology, the OIP probe and logging system are used for petroleum hydrocarbon and electrical conductivity measurements [29]. Based on direct push drilling, hydraulic investigation methods such as DP slug tests, DP injection logging, and the hydraulic profiling tool are applied to rapidly delineate hydrogeological structures [30]. Remote sensing and geophysical methods provide a means of investigating and locating buried waste materials. Among them, geophysical methods provide a means to detect contaminant plumes under certain conditions. In addition, geophysical techniques allow for a non-destructive investigation of shallow ground without causing further anthropogenic spread of contaminants. The electromagnetic induction mapping, combined with direct propulsion in situ measurements, can be used to determine the relationship between the heterogeneous resistivity and contaminant distribution [31].

However, when drilling in non-homogeneous formations, the depth and point position of direct push drilling may be deviated by the force of the formation. Deviations in the depth and point locations of direct push drilling will lead to inaccurate conclusions due to the uneven spatial distribution of contaminants. Therefore, the adaptive adjustment and control of the drilling trajectory is the improvement of the current study. The support project of this study combines in-situ investigations and trajectory control to effectively bridge the gap between existing techniques. Precise drilling and control are the key technologies indicated in Figure 1. In this paper, a chain-type direct push (C-DP) drilling system is introduced, including its structure, composition, and main functions. It can simulate the on-site drilling to detect the drilling control process. The rig can also be equipped with selective semi-permeable membranes, heating equipment, and information acquisition and transmission equipment. In addition, it can provide indoor test conditions for the study of the control of the trajectory of direct push investigation equipment. The C-DP drilling rig is mainly applied in the field of contaminated site investigations, and it does not vibrate and rotate during drilling operation.

## 2. Principle of the Design

The chain drive is relatively easy to install, and it can be readily redesigned and reconfigured in comparison to gear drive systems. In addition, it is widely used in automobiles and transmission machinery as an effective power transmission choice [32]. The chain drive has a higher transmission power and efficiency than the belt drive and a longer transmission distance than the gear drive. In addition, it can avoid the space requirements of hydraulic cylinders. Based on the above characteristics, the chain is selected as the drive for the C-DP drilling rig. The chain drive has a single, duplex, and triplex chain conventional drive forms. The drive characteristic of the triplex chain is shown in Figure 2.

The operating transmission ratio of the triplex chain is shown in Equation (1).
(1)$$i=\frac{n_{1}}{n_{2}}=u=\frac{z_{2}}{z_{1}}$$
where:

*i*—operating transmission ratio,*n*_1_—input speed,*n*_2_—output speed,*u*—kinematic transmission ratio,*z*_1_—teeth number of driving sprocket,*z*_2_—teeth number of driven sprocket.

The efficiency of the triplex chain transmission is usually less than 1/3 of the real load ability, and the single chain requires a larger size to meet the needs of the drive. Based on the consideration of the different chains’ characteristics, the duplex chain is used in the C-DP drilling rig, and the operating transmission ratio is one.

The previous work referred to the tree diagram to design the chain drive system, which can improve the reliability of the chain [33]. The friction chain rig is used at the Transylvania University of Brasov in order to increase the tensioning force and keep the force pushing on the guide [34]. Therefore, the tree diagram method was applied to the design of the C-DP drilling rig, and the rig is equipped with a chain fastening device to adjust the tensioning force.

## 3. General of C-DP Drilling Rig

The modular theory [35] is used to design and produce the C-DP drilling rig, which is composed of the direct propulsion module, hydraulic system, adjustment module, and the operation—control module and stratum—auxiliary module. The proposed C-DP drilling rig for contaminated sites can carry a variety of in-situ testing devices and different pilot bits, which can also perform precise direct push drilling at different angles and deviation correction tests.

### 3.1. C-DP Drilling Rig Description

The overall design of the C-DP drilling rig and the physical drawings are shown in Figure 3. The foundation is used to carry all loads, which is divided into two parts: the gantry and the base. The guide rail is manufactured to secure the gantry and to make the gantry move on it. The derrick acts as the carrier of the chain, and it is secured on the base of the drill rig. The lifting cylinder is connected with the foundation and the derrick using bolting. The console is placed on the gantry at the lower end of the derrick, which is designed to control the drilling rig. The hand chain hoist is fixed in the middle of the derrick by a rotatable bracket, which includes a main arm and an auxiliary arm. The main arm can be rotated within a range of 270°, and the auxiliary arm can be rotated 360° to realize a large range of movement of the soil sample buckets. The C-DP drilling rig is highly mobile and flexible; for example, it can freely adjust the drilling angle and cause little disturbance to the sites during drilling operation.

### 3.2. Direct Propulsion Module

The direct propulsion module is the core module that is made up of the derrick, duplex chain, hydraulic motor, and power head. It is used to carry the loads during the drilling operation. The chain, the movement of which is controlled by the different speeds and steering of the hydraulic motor, provides direct pushing and pulling power for the module. This approach minimizes rig space requirements and increases drilling depth in the single pass compared to conventional hydraulic direct push rigs. In addition, chains as a direct power source for drilling rigs have more forms of monitoring options. The model and the physical drawings of the direct propulsion module are shown in Figure 4.

The rotational speed of the hydraulic motor is controlled by the reducer with a reduction ratio of 20. At the same time, the torque of the hydraulic motor is increased through the reducer, which can provide a greater feeding force. The power head is composed of a core pressure-bearing copper column and fixing gaskets. The power head has a longitudinal penetration core pressure-bearing copper column, and they are fixed by 10 fastening bolts and gaskets. The connection method can improve the stability of the drilling tool power head compared with the threaded connection. The core pressure-bearing copper column is drilled with a 20 mm through hole for the passage of cables, which provides conditions for the real-time acquisition and transmission of drilling parameters. This module can provide direct propulsion for selective semipermeable membranes, heating devices, and information acquisition and transmission devices.

### 3.3. Regulating Module

The module is mainly composed of the foundation, moving cylinder, and lifting cylinder. The model and physical drawings of the lifting cylinder are shown in Figure 5. The linear movement of the drilling position in the range of 100 mm and the adjustment of the drilling angle of 0–90° can be realized by the mobile cylinder and lifting cylinder, respectively. The actions of the hydraulic cylinders are realized through the operation—control module.

### 3.4. Operation—Control Module

The operation—control module is one of the human-computer interaction modules that is used to control the actions of the direct propulsion module and the regulating module. The module includes the pressure gauge, as well as the operating levers and their control valves. The model and physical drawings of the console are shown in Figure 6. The pressure of the hydraulic system is displayed in real time, which can avoid the damage of thr C-DP drilling rig caused by excessive working pressure. The module also can adjust the drilling speed (coarse adjustment/fine adjustment) and angle (0–90°). All adjustments of the module are stepless, allowing continuous changes of input pressure, drilling angle and drilling speed. This approach allows the adjustment of drilling parameters to be more suitable for a wide range of direct push drilling situations.

### 3.5. Stratum—Auxiliary Module

The module is composed of three separate soil sample buckets and a hand chain hoist. Among them, the separate soil sample buckets are made of high-strength alloy steel and equipped with two lugs, respectively. They can prepare contaminated soil layers with a diameter of 700 mm and a height of up to 1100 mm. The hand hoist can control the lifting and movement of the soil sample buckets, which can improve the efficiency of soil layers replacement and shorten the test period.

### 3.6. Hydraulic System

The hydraulic system includes an electric motor, oil tank, gear pump, and oil pipe. The motor is the YE3-160M-4 three-phase asynchronous motor; its main function is to drive the movement of the gear pump and provide energy input for the system. The gear pump can continuously suck the hydraulic oil into the vacuum space to generate stable hydraulic energy with the advantages of small size, lightweight, and strong self-suction. The performance parameters of the motor are shown in Table 1.

The oil tank is designed to store oil and provide an energy transmission medium for the C-DP drilling rig. It ensures that the oil can return to it when the drilling operation stops. In addition, the oil tank also has functions such as heat dissipation and precipitation of impurities.

## 4. Drilling Tests

### 4.1. Test Materials

The constructed homogeneous simulated soil layer is shown in Figure 7, which was mixed 1:1 with reticulate red clay and quartz sand. Reticulate red clay was taken from the foothills of Yuelu Mountain, Changsha, Hunan, China. The physical parameters of the reticulate red clay are shown in Table 2. After reticulate red clay and quartz sand were mixed and compacted, they were made into a simulated soil layer with a height of 900 mm and a diameter of 700 mm. This soil layer is only one of the layers constructed for the indoor direct drilling tests. In addition, the stratum—auxiliary module allows the construction of strata with different physical characteristics to simulate different depths of strata, which can be used to simulate drilling processes at different depths under indoor conditions.

### 4.2. Test Procedure

The test procedure consists of four steps. First, the C-DP drilling rig was adjusted to the state of vertical direct push drilling, as shown in Figure 8, and the inclination angle of the borehole was 0°. Second, the drilling speed adjusting lever was changed to the drilling state. Third, the parameter calibration was performed with input pressures in the range of 1.5–5.0 MPa. Fourth, suitable input pressures were selected for the simulated soil drilling test. Where, in the process of the parameter calibration, the input pressure of the hydraulic motor depends on the test scheme in Figure 9.

## 5. Results and Discussion

### 5.1. Parameters Calibration

For the analysis of the test results in parameters calibration, drilling speeds with different input pressures of hydraulic motor conditions are discussed. The results of the drilling speeds are shown in Figure 10. It can be seen that the real-time drilling speed maintains oscillation stability, and it oscillates mainly in a range of 6 mm/s for different input pressures. In addition, the real-time speed was increased by increasing the input pressure. Figure 10 also shows that, under the action of the higher-value input pressure, the distribution of the real-time drilling speed becomes dispersed. In this paper, the standard deviation is used to characterize the degree of dispersion of the input pressure and real-time drilling speed, as shown in Figure 11. The results show that the dispersion of the input pressure is small, and it proves the drilling rig can provide a stable fluid pressure. As the input pressure increases, the standard deviation of real-time speed rises. Therefore, with the increase in the input pressure, the real-time speed becomes relatively unstable, which may be due to the damage to the internal parts of the hydraulic motor. However, it does not affect the stability of the drilling process.

The average real-time speed increased by increasing the average input pressure, as illustrated in Figure 12. The drilling speed of the C-DP drilling rig can be divided into three stages: high-speed stage, medium-speed stage, and low-speed stage. Equation (2) explains the relationship between the drilling speed and the input pressure in the parameter calibration. The operators can adjust the input pressure of the hydraulic motor to control the drilling speed.
(2)y=2.71⋅x+1.655
where:

*y*—average drilling speed, mm/s.*x*—input pressure of hydraulic motor, MPa.

**Figure 12 ijerph-20-03757-f012:**
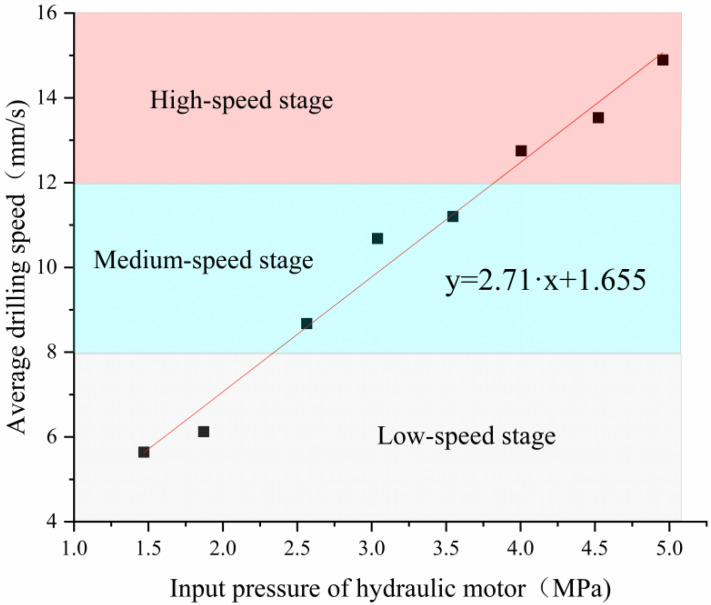
Average drilling speed at different input pressures of hydraulic motor.

### 5.2. Simulated Soil Drilling Test

Based on the results in Section 5.1, the input pressure of the hydraulic motor in this test is 2.000 MPa. However, as shown in Figure 13, the input pressure fluctuates between 1.935 MPa and 2.060 MPa, but its average value is 2.000 MPa. Under the action of this working pressure, the drilling distance is 462.461 mm, the drilling time is 87.545 s, and the average drilling speed is 5.283 mm/s. The average drilling speed is lower than the speed calculated by Equation (2), but the real-time drilling speed is more stable. Therefore, the correction of Equation (2) needs to be performed in future work based on multiple drilling tests in different soil layers. As shown in Figure 13, the inclinations of the drilling tool present step shape fluctuations, and they vary from 0.3° to 0.9°. The phases in which the inclinations remain constant correspond to the phases in which the real-time drilling speed remains relatively stable, which shows that large fluctuations in real-time speed can affect the stability of the drilling tool. Therefore, real-time drilling speed needs to be kept stable to obtain accurate data during the contaminated sites investigation. The drilling effect is shown in Figure 14. It can be seen that the C-DP drilling rig can provide a good hole formation effect and remain stable in the drilling operation.

In order to improve the applicability of this rig in field applications, based on the obvious response in the test, it suggests that the drilling speed should firstly be maintained at 6 mm/s (low-speed stage) for a long period of time. The reason is that during the medium- and high-speed drilling stages, drilling speeds become relatively dispersed, and the fluctuations of drilling speed occur more frequently. For the drilling trajectory in the field, the law of change can be determined according to the real-time drilling speed; when the real-time drilling speed remains constant for a long time, it means that the drilling inclination remains relatively stable to achieve a more accurate direct push drilling process. Therefore, the reasonable selection for the direct push drilling speed is important in the field of direct push drilling. As well as with the duration of the drilling depth, the drilling speed may have a tendency to become smaller. The technical parameters can be optimized by adjusting the input pressure of the hydraulic motor in the process of implementing direct push drilling. The drilling speed and inclination angle evolution in the drilling’s different stages can be obtained from the tests, which can provide the theory support for optimizing the parameters of direct push drilling operation in the field.

## 6. Conclusions

(1)This paper proposed a new chain-type drilling structure design using a chain for drilling contaminated sites. The duplex chain design has a high transmission efficiency, decreases the space occupied, improves the drilling rig’s mobility, and allows for more various sites.(2)The modular theory was used to design drilling rigs, and the C-DP drilling rig can perform direct push drilling tests regarding drilling trajectory. The results show that the direct propulsion module and adjustment module can perform long-term continuous direct push drilling at a set angle. The stratum—auxiliary module can realize the simultaneous preparation of various contaminated soil layers and can be replaced with each other. The hydraulic system can provide power for different modules of the whole drilling rig.(3)A new C-DP drilling rig prototype was designed and built to validate our models while drilling in the indoor simulated soil layers.(4)The parameter calibration results show that the C-DP drilling rig can provide liquid pressure from 0 to 5 MPa. Moreover, the relationship between the drilling speed and input pressure is obtained. The simulated soil drilling test results show that the C-DP drilling rig can achieve accurate direct push drilling in a vertical state when the input pressure of the hydraulic motor is 2.000 MPa and the single pass is 462.461 mm. In addition, no large-angle deflection occurs under the condition of homogeneously soil layers.

The principles developed in this paper can be used to design and investigate C-DP drilling rigs for the research and development of integrated drilling and investigating technology and equipment.

## Figures and Tables

**Figure 1 ijerph-20-03757-f001:**
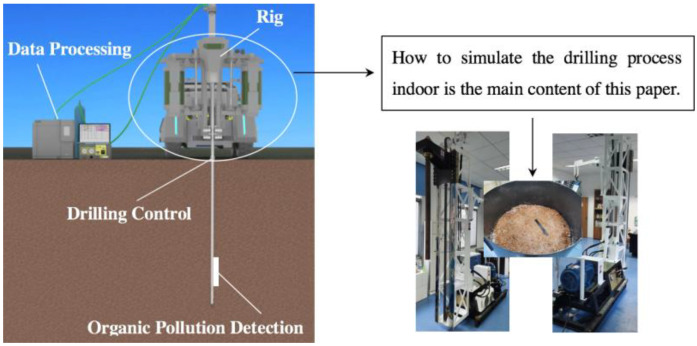
Drilling Control.

**Figure 2 ijerph-20-03757-f002:**
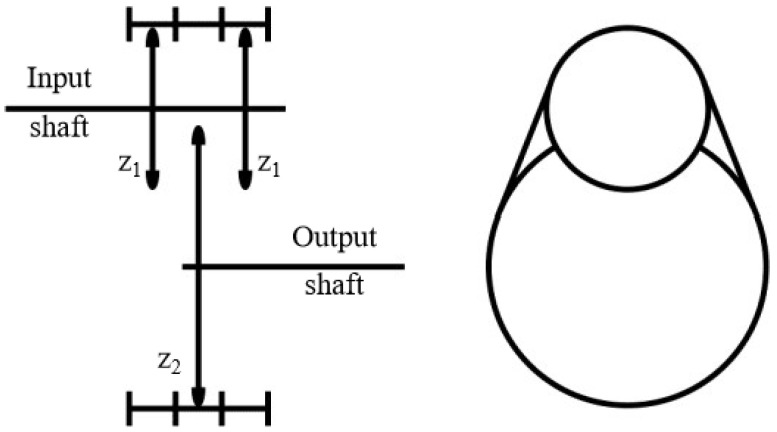
Characteristic of triplex chain transmission.

**Figure 3 ijerph-20-03757-f003:**
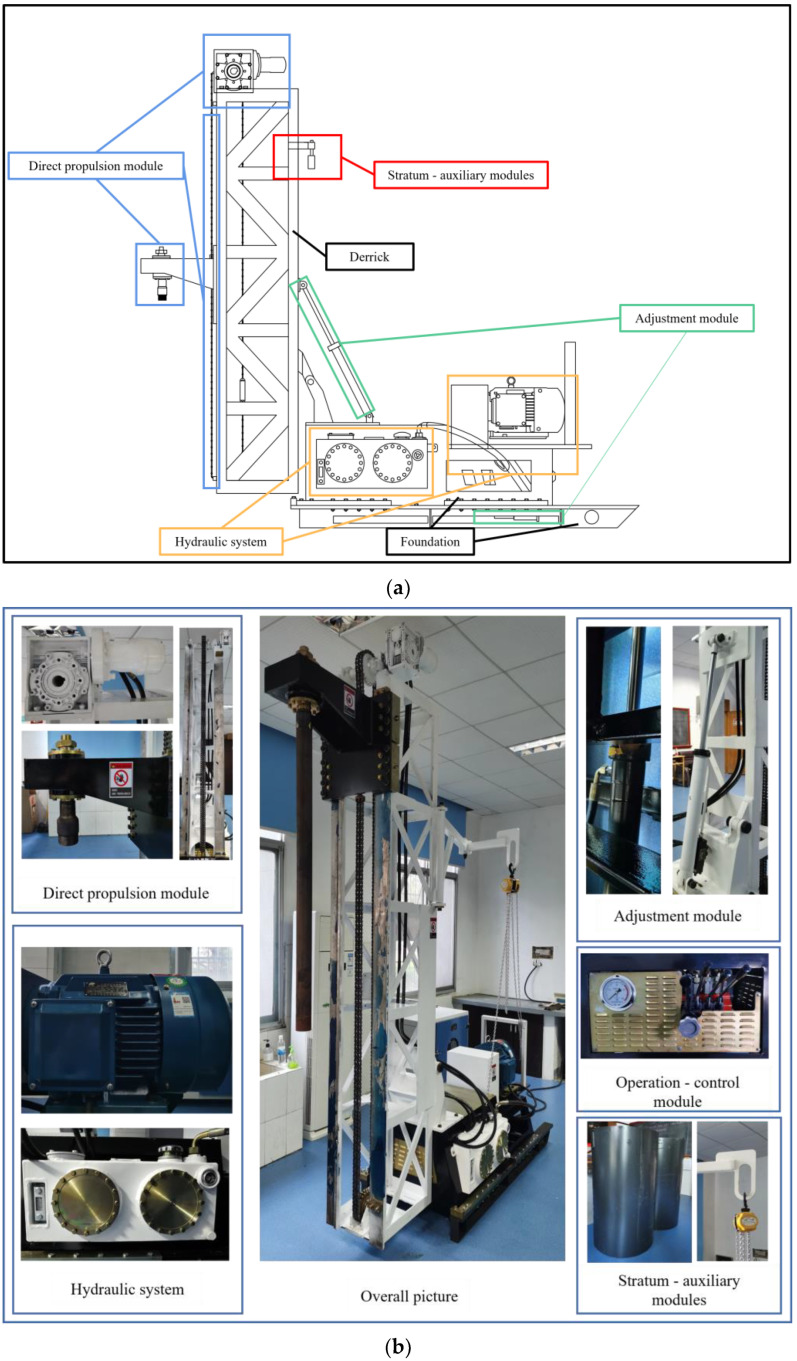
Overall design and physical drawing of the C-DP drilling rig. (**a**) Model drawing; (**b**) Physical drawing.

**Figure 4 ijerph-20-03757-f004:**
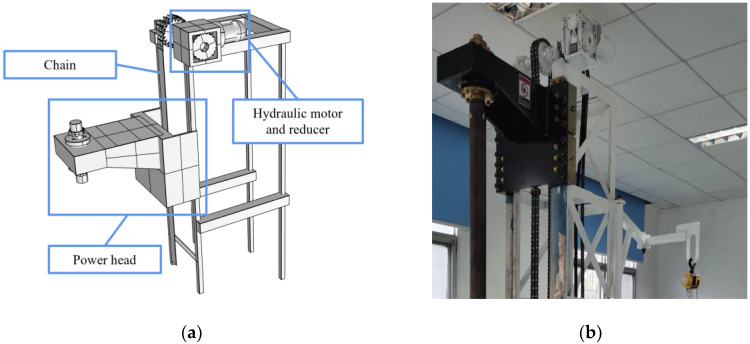
Model and physical drawing of the direct propulsion module. (**a**) Model drawing; (**b**) Physical drawing.

**Figure 5 ijerph-20-03757-f005:**
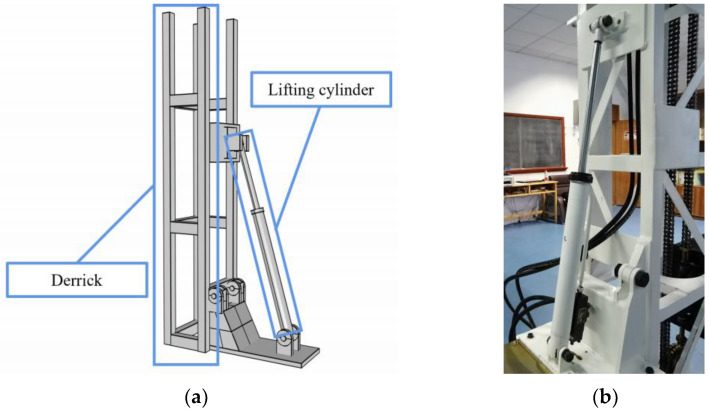
Model and physical drawing of the lifting cylinder. (**a**) Model drawing; (**b**) Physical drawing.

**Figure 6 ijerph-20-03757-f006:**
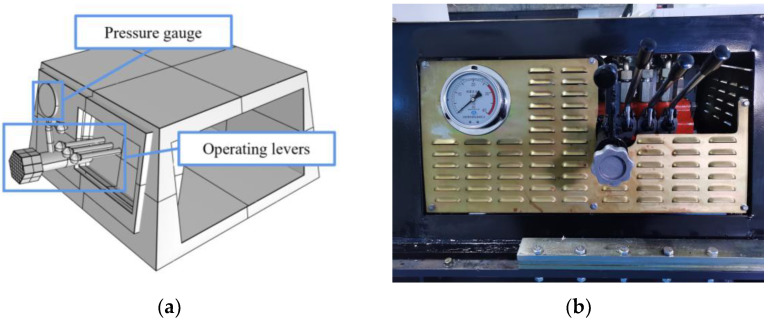
Model and physical drawing of the console. (**a**) Model drawing; (**b**) Physical drawing.

**Figure 7 ijerph-20-03757-f007:**
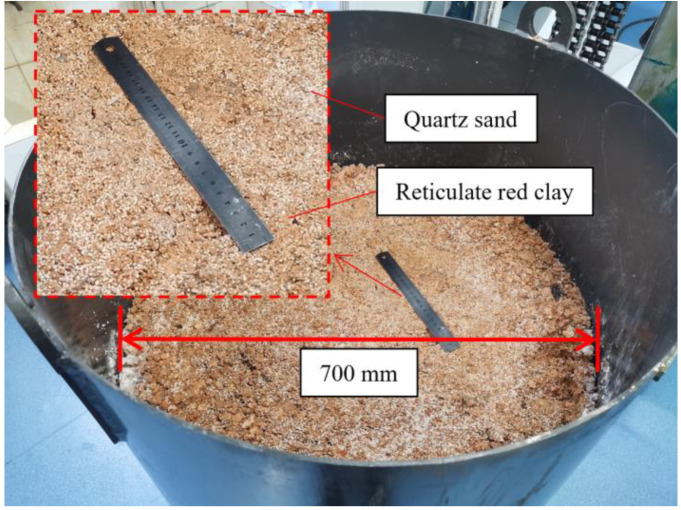
Simulated contaminated soil layer.

**Figure 8 ijerph-20-03757-f008:**
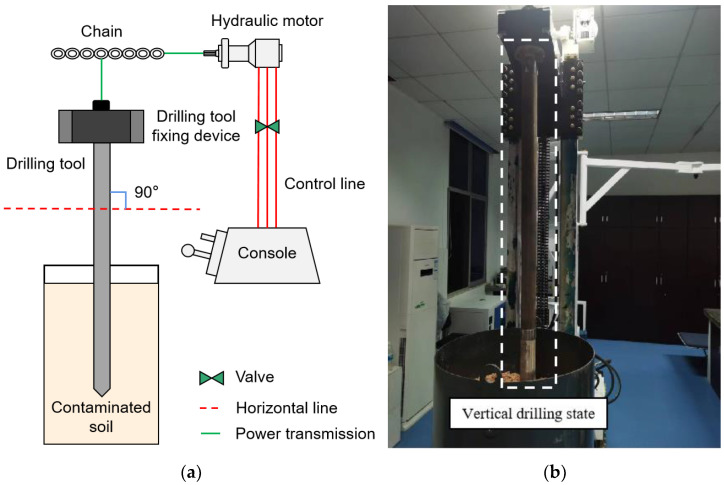
Working state of the C-DP drilling rig. (**a**) Flow Chart; (**b**) Physical drawing.

**Figure 9 ijerph-20-03757-f009:**
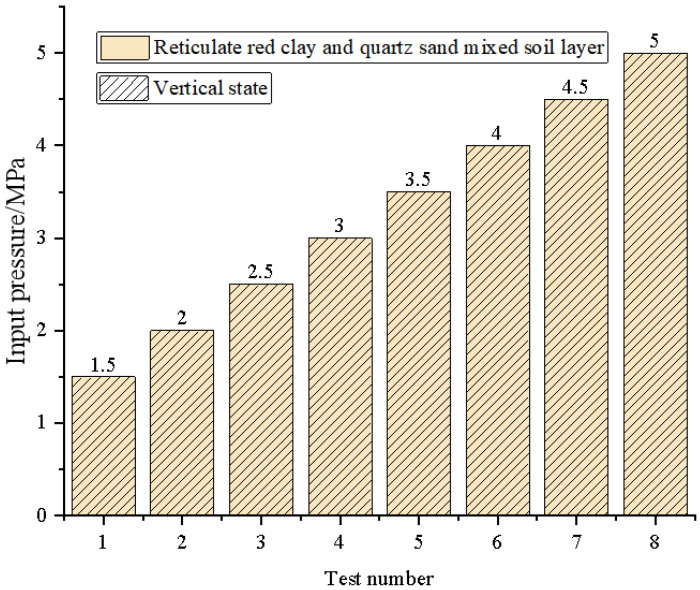
Test scheme.

**Figure 10 ijerph-20-03757-f010:**
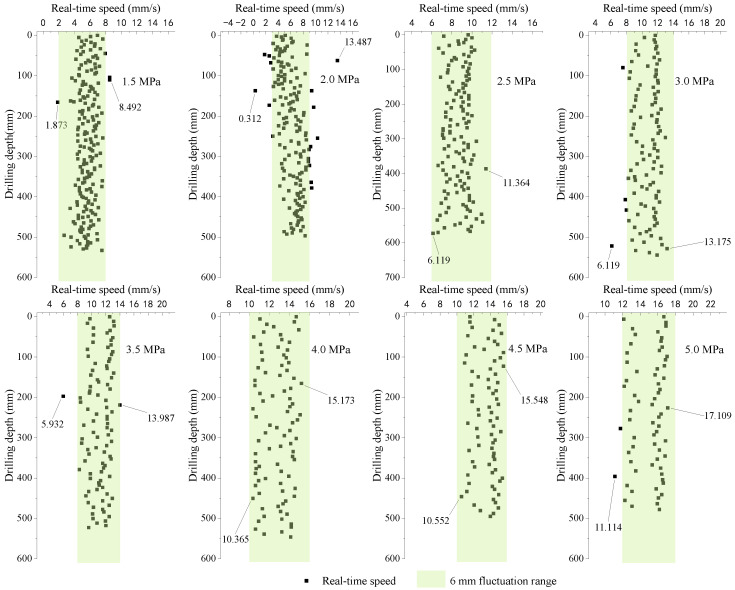
Real-time speed variation.

**Figure 11 ijerph-20-03757-f011:**
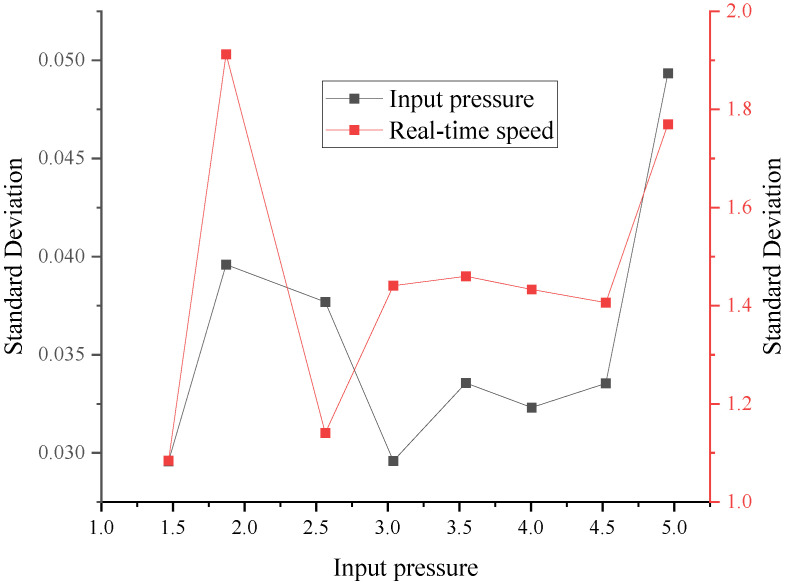
Standard deviation variation.

**Figure 13 ijerph-20-03757-f013:**
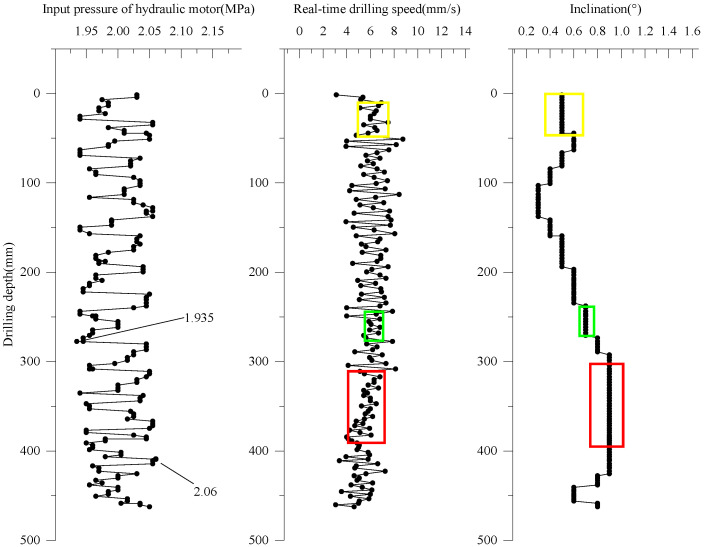
Change of drilling parameters during drilling process.

**Figure 14 ijerph-20-03757-f014:**
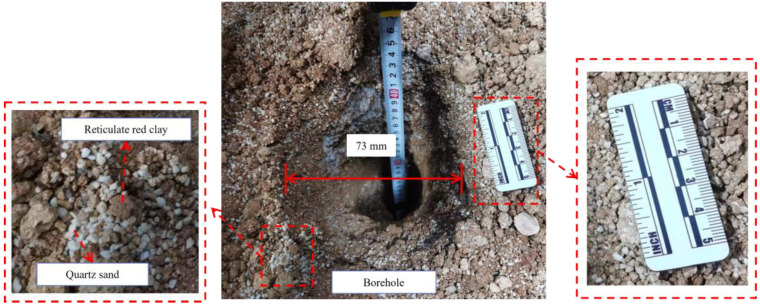
Direct push drilling effect of the C-DP drilling rig.

**Table 1 ijerph-20-03757-t001:** Motor performance parameters.

Performance	Parameter
Rated Power (kW)	11
Rated Voltage (V)	380
Rated Current (A)	21.5
Rated Speed (r/min)	1460
Rated Efficiency (%)	91.4
Rated Frequency (Hz)	50
Protection Class	IP55
Insulation Class	Class F

**Table 2 ijerph-20-03757-t002:** Physical parameters of the clay (Data from [36,37]).

Properties	Value
Water content (%)	24.3
Density (g/cm^3^)	1.9
Specific gravity	2.71
Effective size (mm)	0.047
Control size (mm)	0.694

## Data Availability

The data supporting reported results can be obtained from the corresponding author upon request.

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
