# Peer review of "Design and Implementation of a Chain-Type Direct Push Drilling Rig for Contaminated Sites"

_ijerph, 2023, doi:10.3390/ijerph20043757_

Round 1
Reviewer 1 Report
The manuscript is well-written and nicely organized. Minor language corrections are needed.
Reviewer 2 Report
Dear authors,
the manuscript in its recent form is a pure technical description of a developed direct push rig, which is indeed interesting. The development is at the stage of a prototype that is tested in the lab. There is no field data or any example for the relevance for in situ sensing with the help of this rig. It’s not clear to me, which kind of sensing tool will be applied with the new rig in order to explore contaminated sites. Therefore the choice of the journal is not clear to me.
The authors state at the end of the introduction that there is an issue with the recent direct push-rigs in terms of disturbance by sampling, excavation or drilling fluid. Here I strongly disagree, as all rigs mentioned by the authors are able to perform in situ-sensing/imaging. There is no drilling fluid and yes, there is an option for soil sampling, but which is only one option. An evaluation of the huge family of in situ-sensing or imaging tools is missing, e.g. Geoprobe MIP system, Dakota Technologies UVOST system. Furthermore there is CPT and in situ hydraulic tests (HPT …) that are applied for a long while for characterization and monitoring of contaminated sites.
To summarize, the technological description might be interesting for another journal. For a manuscript with a focus on potential application on contaminated sites the research, motivation and most important the described state-of-the-art is not appropriate. I'm not able to recommend this manuscript for this journal and need to reject it.
Köber, R., Hornbruch, G., Leven, C., Tischer, L., Großmann, J., Dietrich, P., Weiß, H., Dahmke, A., (2009): Evaluation of combined direct-push methods used for aquifer model generation, Ground Water 47 (4), 536 – 546
Cassiani, G., Binley, A., Kemna, A., Wehrer, M., Flores Orozco, A., Deiana, R., Boaga, J., Rossi, M., Dietrich, P., Werban, U., Zschornack, L., Godio, A., JafarGandomi, A., Deidda, G.P., (2014): Noninvasive characterization of the Trecate (Italy) crude-oil contaminated site: links between contamination and geophysical signals, Environ. Sci. Pollut. Res. 21 (15), 8914 – 8931
Davidson, K. B., Lake, C. B., Sweet, B., & Spooner, I. S. (2021). Examining the ultraviolet optical screening tool as a viable means for delineating a contaminated organic sediment. Science of the Total Environment, 799, 149408.
Reviewer 3 Report
The effort the authors put on to prepare this manuscript is commendable. However, the authors need to provide more clarification on the following items;
1. What makes this system better, a comparison with existing systems needed
2. Please review the language on the paper and language need improvement
3. The testing lacks details needed for field application and testing up to 600mm is insufficient.
4. There are not sampling pictures of samples tested and compared to indicate the system is better than existing systems
Round 2
Reviewer 2 Report
Dear authors,
thanks for your response. However I have to admit that I can not follow your argumentation concerning HPT, MIP etc. in line 92-95: "However, the above approach is based on the separation of drilling and investigation, that is, direct push drilling is completed and then the investigation tools are lower into the borehole. The time interval between the use of direct push drilling rig and investigation tools can lead to volatilization of VOCs, which can result in inaccurate survey results. " From my point of view this is simply not true. The family of in situ sensing tools is not separating drilling and sensing. It's actually the innovation that sensing is performed while pushing/hammering the sensors into the ground. As you state this is applied since the 1990s.
I cannot accept the paper as long as you start from a wrong hypothesis and don't describe the recent state of the art.
